# Investigating the Effects of Donors and Alkyne Spacer on the Properties of Donor-Acceptor-Donor Xanthene-Based Dyes

**DOI:** 10.3390/molecules28134929

**Published:** 2023-06-22

**Authors:** Ishanka N. Rajapaksha, Jing Wang, Jerzy Leszczynski, Colleen N. Scott

**Affiliations:** 1Department of Chemistry, Mississippi State University, Mississippi State, MS 39762, USA; inr10@msstate.edu; 2Department of Chemistry, Physics and Atmospheric Sciences, Jackson State University, Jackson, MS 39217, USA; jingw@icnanotox.org (J.W.); jerzy@icnanotox.org (J.L.)

**Keywords:** donor-acceptor-donor, NIR dyes, xanthene dyes, amine donors, alkyne spacers

## Abstract

NIR dyes have become popular for many applications, including biosensing and imaging. For this reason, the molecular switch mechanism of the xanthene dyes makes them useful for in vivo detection and imaging of bioanalytes. Our group has been designing NIR xanthene-based dyes by the donor-acceptor-donor approach; however, the equilibrium between their opened and closed forms varies depending on the donors and spacer. We synthesized donor-acceptor-donor NIR xanthene-based dyes with an alkyne spacer via the Sonogashira coupling reaction to investigate the effects of the alkyne spacer and the donors on the maximum absorption wavelength and the molecular switching (ring opening) process of the dyes. We evaluated the strength and nature of the donors and the presence and absence of the alkyne spacer on the properties of the dyes. It was shown that the alkyne spacer extended the conjugation of the dyes, leading to absorption wavelengths of longer values compared with the dyes without the alkyne group. In addition, strong charge transfer donors shifted the absorption wavelength towards the NIR region, while donors with strong π-donation resulted in xanthene dyes with a smaller equilibrium constant. DFT/TDDFT calculations corroborated the experimental data in most of the cases. **Dye 2** containing the N,N-dimethylaniline group gave contrary results and is being further investigated.

## 1. Introduction

Organic dyes are of significant interest for many applications such as light-emitting diodes [1,2,3], organic photovoltaics [4,5], dye-sensitized solar cells [6,7,8,9], photodynamic therapy [10,11,12], and biological sensing and imaging [13,14,15]. Metal-free organic dyes are specifically desirable because they are usually inexpensive, lightweight, and easy to process. The classification of these dyes is based on their core-structures such as squaraines [16,17], cyanines [18,19,20,21], boron difluoride dipyrromethene (BODIPY analogs) [22,23], diketopyrrolopyrroles (DPP) [24,25,26], and xanthene derivatives [27,28,29]. Among these dyes, xanthene-based dyes are desirable for biological sensing and imaging due to their unique molecular switching mechanism, excellent photophysical properties in the opened form, and extremely short synthetic route. Fluorescein and rhodamine are the two most common xanthene dyes and are widely used for cellular imaging and sensory applications due to their high fluorescence quantum yield, large molar absorptivity coefficient, high photostability, and water solubility [28,30]. Fluorescein and rhodamine dyes can switch from a colorless spirocyclic structure (closed form) to a colorful emissive form (opened form) upon stimulation by analytes [30]. Typically, these dyes have absorption and emission wavelengths around 400–600 nm in the visible region. This property allows them to be used as colorimetric probes. Recently, there has been interest in our group and others in extending the absorption and emission wavelengths into the far-red, near-infrared (NIR), and shortwave infrared (SWIR) regions for sensing and imaging applications in biological tissues. Many strategies have been used to modify their structures to obtain far-red to NIR dyes. These strategies include replacing the bridged oxygen atom with Si [31,32,33], P [34,35,36], or C [33,37,38], extending the conjugation through an alkene group [39], and using the donor-acceptor-donor approach [40,41,42]. For example, Lavis and coworkers reported the synthesis of carbofluorescein and carborhodamine by substituting the bridged oxygen atom with a quaternary carbon group [38]. The new dyes exhibited a 50 nm bathochromic shift in the absorption and emission spectra compared with their parent xanthene dyes. Fu et al. discovered the siliaanthracene derivative (TMDHS, 2,7-N, N, N′, N′-tetramethyl-9-dimethyl-10-hydro-9-siliaanthracene) by replacing the bridged oxygen atom with silicon [31]. The new compound, TMDHS, exhibited a 90 nm bathochromic shift in its absorption and emission wavelengths compared with the parent rhodamine compound. Nagano and coworkers further explored rhodamine derivatives by replacing the bridging oxygen atom with group 14 elements Si, Ge, and Sn [43]. Si- and Ge-containing rhodamine derivatives exhibited absorption and emission maxima in the far-red to NIR I region; however, the Sn derivative was not studied due to its instability in the solution. Our group also explored silafluorescein compounds as HOCl probes [44]. The dye also yielded maximum absorption (586 nm) and emission (606 nm) levels that were 90 nm red-shifted compared with the original fluorescein. Yamaguchi et al. discovered highly photostable rhodamine and fluorescein derivatives by replacing the bridging oxygen atom with a phosphine oxide moiety. The phospha-fluorescein dye (POF) showed absorption and emission at 627 nm and 656 nm, while the phospha-rhodamine dyes (POR) showed maximum absorption and emission at 712 nm and 740 nm, respectively [35]. Additionally, Yuan et al. reported a unique class of NIR fluorescent dyes that can transform between fluorescence “on” and “off” modes with the switching of the lactone ring [39]. This new class of xanthene-dyes have extended π-conjugation through the xanthene core and exhibited absorption maxima between 688 and 728 nm and emission maxima between 721 and 763 nm with excellent photophysical properties. However, due to the lengthy synthetic steps that are required to obtain many of these NIR I xanthene-based dyes, they are not desirable for scale-up protocols.

In the last few years, our group has been investigating a donor-acceptor-donor (D-A-D) approach that is based on charge transfer between the donor and acceptor molecules to obtain far-red to NIR xanthene-based dyes. One of the key aspects of our approach is the short and straightforward synthetic routes to the dyes. For example, pyrrole and indole were coupled at the C-2 position of the xanthene core using the Suzuki–Miyaura reaction to prepare the dyes in two steps [40], while C-H activation chemistry was used to couple indolizine at the C-3 position to the xanthene core [42]. The xanthene dyes with pyrrole and indole donors resulted in maximum absorption and emission wavelength between 650 and 775 nm, whereas the indolizine donor resulted in maximum absorption and emission in the shortwave infrared (SWIR) region of 900–1100 nm. Recently, we incorporated a thiophene spacer between the xanthene acceptor and the amine donors. It was shown that the combination of the amine and thiophene acts as a strong donor, while the thiophene spacer also acts to extend the conjugation. With this approach, we prepared two types of dyes: a thienylpiperidine donor (**CR880**) [45], and a series of thienyldibenzazepine donors (**SCR-1**–**SCR-3**) [46]. Moreover, the SCR dyes had absorption and emission maxima in the SWIR region (890–1260 nm), and **SCR-1** was used in a nano-receptor (**SCR-NO**) to detect nitric oxide (NO) in a mouse liver model that was induced to produce NO. However, due to the low quantum yield of the fluorescent signal, a photoacoustic imaging technique was used to obtain the images. In addition to our group, Liu et al. recently reported a new set of xanthene derivatives containing a styryl spacer with oxygen and nitrogen-based donors (**VIX-1**–**VIX-4**) that resulted in photophysical properties in the NIR I to SWIR window (720–1260 nm) [47]. Furthermore, **VIX-4** was used to effectively demonstrate in vivo imaging of blood circulation.

With our success in developing several NIR xanthene-based dyes using the D-A-D approach, it was still left to determine the importance of the choice of donors that would allow for the best photophysical properties of the dyes, particularly in the opened form. For this reason, we evaluated a series of dyes containing different donors and an alkyne spacer. The donors and alkyne spacer were chosen to compare our dyes with the previously published VIX dyes containing similar donors and an alkene spacer [47]. Our studies combine experimental and computational analyses. It has been demonstrated that a well-balanced combination of experimental and computational methodology results in a more in-depth understanding of the molecular structures of compounds and their properties. As such, our group has been applying computational methods in combination with the experimental data to study structures and properties of various dyes. These studies have resulted in the design of novel dyes with important characteristics for efficient light harvesting materials [48]. In this manuscript, we report a series of xanthene-based D-A-D dyes with an alkyne spacer between oxygen and nitrogen-based donors (Figure 1). The dyes were synthesized in short synthetic routes and the effects of the donors and alkyne spacer on their photophysical properties were studied spectroscopically and with DFT/TDDFT calculations. The knowledge gained from the most successful computational approaches developed in the previous studies are applied in the current project. Herein, we report our results.

## 2. Results and Discussions

The design of the new dyes is based on selecting common oxygen and nitrogen-based donors (**D-1** to **D-4**–anisole, N,N-dimethylaniline, N,N-diphenylaniline, and carbazole, respectively) and connecting them to the xanthene core through an alkyne spacer. The synthesis of these new xanthene-based dyes began by first coupling the different donors containing the alkyne moiety to 3′,6′-dibromofluoran using the Sonogashira coupling reaction according to a previously reported procedure (Figure 1) [49,50]. It should be noted that 4-ethynylanisole, 4-ethynyl-N,N-dimethylaniline, and 4-ethynyl-N,N-diphenylaniline were commercially available; however, the other ethynyl donor needed to be prepared according to procedures in the literature [51,52]. Subsequently, 9-phenylcarbazole was coupled to ethynyltrimethylsilane, followed by removal of the trimethylsilyl group to give the alkyne precursor (Scheme S1). All the alkyne compounds were coupled to the xanthene core according to the conditions of Figure 1.

Following the synthesis of the dyes in their closed forms, we attempted to convert them to their ethyl ester based on our previous work [40]. Unfortunately, we were unsuccessful in the synthesis of the ethyl esters. Consequently, we treated the dyes with trifluoroacetic acid (TFA) to access the carboxylic acid opened form (Appendix A). TFA in dichloromethane (DCM) was added to the dyes and the absorption spectra were obtained 30 min later. The absorption spectra for **Dyes 1**–**4** are shown in Figure 1a. The maximum absorption wavelength (abs λ_max_) for **Dye 1** was obtained at 630 nm. While alkoxy groups are known to be strong donors, amines are usually stronger donors [53]; consequently, **Dyes 2**–**4** were expected to have a stronger charge transfer interaction with the xanthene core, leading to a lower bandgap. In fact, the strength of aromatic amine donors was studied by Marder and coworkers who compared the intermolecular electron transfer (ET) process of the aromatic amines [54]. The HOMO energy levels and ionization potentials were used to determine the ET donor strength of the amines, which was expected to affect the abs λ_max_. It was determined that the order of the ET donor strength was **D3** > **D4** > **D2**. While **D1** was not included in the study, it was expected to be less donating than the amine donors. We were pleased to see that the experimental abs λ_max_ followed this ET donor strength. For example, the N,N-diphenylaniline donor (**Dye 3**) resulted in the longest abs λ_max_ (850 nm), followed by **Dye 4** and **Dye 2**. Surprisingly, the N,N-dimethylaniline dye (**Dye 2**) possessed a shorter abs λ_max_ (515 nm) than **Dye 1** (630 nm). Additionally, the 9-phenylcarbazole donor (**Dye 4**) exhibited two prominent absorbance peaks with equal intensities: the first at 495 nm representing the π-π* transition, and the second at 675 nm from the charge transfer transition (Figure 1a). We hypothesized that these two prominent transitions are due to rotational barriers caused by the rigidity of the fused carbazole donor, leading to structural differences. This rotational barrier could be attributed to the large twist angle between the phenyl ring and the carbazole, which was calculated to be around 60°, resulting in a twisted or separated π-system. Consequently, the absorption occurred from different parts of the of the molecule. It is worth noting that the abs λ_max_ of **Dyes 1** and **2** were blue-shifted compared with those of the analogous xanthene **VIX** dyes reported previously [47]. **Dye 1** is the analogous xanthene dye for **VIX-2**, which exhibited absorbance at 653 nm, a 20 nm red-shift compared with **Dye 1**. On the other hand, **VIX-3** with the N,N-dimethylaniline group showed excellent photophysical properties in DCM with absorbance at 877 nm, whereas **Dye 2** showed absorbance at 515 nm.

The absorption spectrum for each dye in the opened form was investigated by density functional theory (DFT) using the M06-2X functional and 6-311G(d,p) basis set. The molecular geometries of all the species considered were fully optimized (Appendix A). The calculations were performed in the presence of TFA and DCM solvents. Based on the TDDFT studies, **Dye 2** has the longest wavelength followed by **Dye 3**, **Dye 1**, and **Dye 4** (Figure 1b). We noticed one discrepancy in the trend in the absorption wavelength between the computational results and the experimental data. **Dye 2** was shown to have the longest wavelength in the calculation, but the shortest among the pool of dyes studied experimentally. The trend in the predicted absorption wavelength of the other dyes falls in the same order as the experimental data. Additionally, the calculation did not depict the two distinct peaks of **Dye 4** that were seen in the experimental data. The intriguing differences between the experimental data and the calculated results for **Dye 2** warrant further investigation involving a careful study with further calculations for the parent compound with other donors. Such studies have been initiated and will be discussed in our future manuscript.

Our inability to convert the dyes to their respective ethyl ester prompts us to investigate the equilibrium between the opened and closed forms. DFT was used to investigate the ground state energies of the closed and opened forms of the dyes, and their stabilities are compared with their π-donor strengths. It was shown that ET-donor and π-donor strengths do not necessarily correlate with each other [54], and the π-donor strength follows the order of **D2** > **D3** > **D4**. In this case, the N,N-diphenylaniline and the 9-phenylcarbazole donors are the weakest π-donors of the amine series. This result can be explained by the steric effects between the phenyl ring and the plane of the nitrogen lone pairs for both **D3** and **D4** with a dihedral angle of 40° [54]. In the case of **D2**, the dihedral angle is closer to 0°. While N,N-diphenylaniline is a stronger ET donor than N,N-dimethylaniline, the latter is a stronger π-donor. A stronger π-donor should lead to a more stable opened form of the dye with a larger equilibrium constant than the weaker π-donors. To test this phenomenon, we calculated the change in the Gibbs free energy (ΔG) for the two forms of each dye in the presence of one equivalent of TFA in DCM as the solvent (Table 1). The calculations were performed at the M06-2X/6-311G(d,p) level with the PCM model. It was determined from the calculation that the closed forms of all the dyes are more stable than the opened forms, as seen from the positive change in enthalpy (ΔH) and the ΔG. Based on the ΔG values, the equilibrium constants (K_eq_) were calculated at 298 K to give 2.66 × 10^−14^, 2.77 × 10^−16^, 1.78 × 10^−15^, and 1.98 × 10^−16^ for **Dyes 1**–**4**, respectively. Based on the equilibrium constants, **Dye 1** has the largest K value and would be more stable in the open form compared with **Dyes 2, 3,** and **4**. This result corroborates the fact that the methoxy group is a strong π-donor that forms a more planar compound. Additionally, the calculation shows that **Dyes 2** and **4** have a smaller K_eq_ value than **Dye 3**, suggesting that **Dye 3** should open more readily than **Dyes 2** and **4**. Interestingly, the calculation predicts that the N,N-dimethylaniline donor has the smallest K_eq_ value and would be the least favorable in the open form, although it was found to be the best π-donor of the amines.

As predicted by the DFT calculation, the equilibrium between the opened and closed forms of the dyes was unfavorable towards the opened form; consequently, a large excess of TFA was required to open the dyes within 30 min (Appendix A). For **Dye 2**, even after 1000 equivalents of TFA was added, only a small amount of the opened form was observed based on the low absorbance intensity of the major peak in the visible region in comparison with the other dyes. On the other hand, **Dyes 1**, **3** and **4**, recorded higher absorbance intensities after 500 equivalents of TFA, suggesting that the equilibrium constant lies more towards the opened form at this stage (Figure 2). To determine if the dyes are at equilibrium after 30 min, we extended the time to 24 h to see if there were significant changes in the absorbance intensities. For **Dyes 1**, **3**, and **4**, the absorbance intensity decreased after 24 h, suggesting that the equilibrium was not yet established after 30 min. However, the absorbance intensity did not change substantially for **Dye 2** after 24 h. Since the absorbance intensities did not increase after 14 h, these results further confirm that the dyes prefer to be in the closed form (Appendix A). To measure the equilibrium constant of the opened and closed forms of the dyes, it is necessary to know the molar extinction coefficient of the dyes in the opened form [55]. Since we were unable to access the fully opened form of the dyes, we are unable to obtain this data. Consequently, we resorted to calculating the change in the absorbance intensity (Δ abs) at the maximum absorbance wavelength for the largest curve in the visible to NIR regions (Appendix A). It was found that the Δ abs after 24 h for **Dye 1** was very large (503) compared with that of **Dye 2**, which was very small (8), and **Dyes 3** (60) and **4** (55), which were comparable with each other. These numbers are not representative of the equilibrium constant; however, they give us an estimation of the relative extent to which the dyes are in the opened form. While the absorbance intensity is related to the extinction coefficient, it is our assumption that all the dyes have relatively similar extinction coefficients. These experimental results follow the π-donor strengths, except for the unusual case of **Dye 2**. Additionally, the results closely follow the DFT prediction of K_eq_. Ongoing experiments are underway to determine the reason for the difference in the trend in π-donor strength and the stability of **Dye 2** in the opened form.

In order to determine the effect of the alkyne spacer on the conjugation of the dyes, we prepared the analogous xanthene-based dyes without the alkyne spacer. **Dyes 6**–**8** were synthesized using the Suzuki–Miyaura coupling reaction, as shown in Figure 2. Unfortunately, we were unsuccessful in the synthesis of the carbazole compound under the reaction conditions. However, since the goal was to determine if the alkyne spacer had a significant effect on extending the conjugation of the dyes, we were satisfied to examine the three dyes synthesized. The absorption spectra for **Dyes 6**–**8** in DCM were taken 30 min after the addition of different equivalents of TFA (Appendix A). The absorption maxima for the dyes without the alkyne spacer were blue-shifted compared with their analogous counterparts (**Dyes 1**–**3**). For example, **Dye 6** had an abs λ_max_ at 582 nm, which is 50 nm less than **Dye 1**, while **Dyes 7** and **8** were 55 and 45 nm blue-shifted compared with **Dyes 2** and **3** (Figure 3). These results suggest that the alkyne spacer increased the conjugation length of the dyes. Finally, we investigated the ease of converting **Dyes 6**–**8** to the opened form. Interestingly, **Dyes 6** and **8** required fewer equivalents of TFA to go to the opened form compared with their alkyne analogs. In fact, **Dye 8** had a higher absorbance intensity after only 50 equivalents of TFA than its counterpart with 100 or more TFA equivalents. Unfortunately, **Dye 7** had an even lower absorbance intensity than its alkyne analog (**Dye 2**). The N,N-Dimethylaniline donor seems to behave differently than the other donors and further study is needed to understand its behavior.

## 3. Materials and Methods

All reagents including catalysts and ligands were obtained from Sigma-Aldrich, Oakwood, Ambeed, or Fisher Scientific. Carbazole and 9-bromo-2,3,6,7-tetrahydro-1*H*,5*H*-pyrido[3,2,1-*ij*]quinoline (Compound **3**) were purchased and used without further purifications. HPLC grade solvents were used for spectroscopic analysis. ^1^H NMR and ^13^C NMR spectra were recorded in deuterated solvents on a Bruker AVANCE 500 Hz spectrometer. *J* values are expressed in Hz, and the quoted chemical shifts are in ppm downfield from the tetramethylsilane (TMS) reference using the residual protonated solvents as an internal standard. The signals have been designated as follows: s (singlet), d (doublet), t (triplet), dd (doublet of doublets), and m (multiplets). High-resolution mass spectra (HRMS) were determined on a Bruker-micrOTOF-Q II Mass Spectrometer. Absorption spectra were acquired using an Agilent Cary 60 UV–Vis instrument.

Synthesis

Compounds 3’,6’-dibromofluoran, xanthene ditriflate, and compound **2** were prepared according to procedures in the literature [50,51,56]. Compound **4**, **D4**, and **D5** were prepared using modified procedures from the literature [52,57].




**Synthesis of 9-(4-ethynylphenyl)-9H-carbazole (D4)**


9-(4-bromophenyl)-9*H*-carbazole [51] (Compound **2**) (0.3104 mmol, 100 mg), PdCl_2_(PPh_3_)_2_ (0.0310 mmol, 21.8 mg), and CuI (0.0310 mmol, 5.90 mg) were added into a dried microwave vial in the glovebox. THF (0.5 mL), diisopropylamine (1.5 mL), and TMS-acetylene (1.552 mmol, 0.23 mL) were added to the vial under nitrogen atmosphere, and the reaction mixture was stirred at 80 °C for 24 h. The reaction mixture was allowed to cool to room temperature and diluted with DI water (30 mL) and DCM (20 mL). The layers were separated, and the aqueous phase was extracted with DCM (3 × 20 mL). The combined organic layer was washed with 0.1 M HCl, saturated with sodium bicarbonate solution, dried over anhydrous sodium sulfate, filtered, and concentrated under reduced vacuum. The crude was purified by flash column chromatography using hexanes as the eluent to give the 9-(4-((trimethylsilyl)ethynyl)phenyl)-9H-carbazole product as a white solid at 95% yield. **^1^H NMR** (500 MHz, CDCl_3_) δ 8.17 (d, *J* = 5.6 Hz, 2H), 7.73 (d, *J* = 8.5 Hz, 2H), 7.55 (d, *J* = 8.4 Hz, 2H), 7.47–7.41 (m, 4H), 7.33 (m, *J* = 5.2, 2.5 Hz, 2H), 0.34 (s, 9H).

9-(4-((trimethylsilyl)ethynyl)phenyl)-9H-carbazole (0.2945 mmol, 100 mg) and THF (2 mL) were added to a dried round bottom flask. To the reaction mixture was added tetra-n-butylammonium fluoride (TBAF) (1M in THF, 0.3534 mmol) under nitrogen atmosphere for 2 h. The reaction mixture was diluted with DI water 30 mL and DCM 20 mL. The layers were separated, and the aqueous phase was extracted with DCM (3 × 20 mL). The combined organic layer was washed with 0.1 M HCl and saturated sodium bicarbonate solution, dried over anhydrous sodium sulfate, filtered, and concentrated under reduced pressure. The crude was purified by flash column chromatography using hexanes as the eluent **D4** as a white solid in 78% yield. **^1^H NMR:** (500 MHz, CDCl_3_) δ 8.14 (d, *J* = 7.8 Hz, 2H), 7.73 (d, *J* = 8.5 Hz, 2H), 7.56 (d, *J* = 8.5 Hz, 2H), 7.42 (d, *J* = 2.4 Hz, 4H), 7.34–7.29 (m, 2H), 3.18 (s, 1H).

Compounds **Dyes 1**–**4** were synthesized by modifying a previously published procedure [49].


**General procedure for synthesis of Dyes 1–4**


3’,6’-Dibromofluoran (1 equiv., 100 mg), alkyne (2.4 equiv.), Pd(PPh_3_)_4_ (2 mol %), and CuI (4 mol %) were added to a dried microwave vial and the mixture was suspended in a nitrogen-saturated piperidine/DMF solution (piperidine 0.2 mL: DMF 1 mL). Then the mixture was heated at 100 °C for 12 h. The mixture was diluted with DCM and washed with brine.


**3′,6′-bis((4-methoxyphenyl)ethynyl)-3H-spiro[isobenzofuran-1,9’-xanthen]-3-one (Dye 1)**


The product was dried over anhydrous sodium sulfate, filtered, and concentrated under reduced pressure. The crude was loaded onto silica gel and purified by column chromatography using Hexanes: ethyl acetate (65:35) mixture to give a yellow-colored solid at 83% yield. **^1^H NMR:** (300 MHz, CDCl_3_) δ ^1^H NMR (300 MHz, CDCl_3_) δ 8.05 (d, *J* = 6.8 Hz, 1H), 7.72–7.61 (m, 2H), 7.54–7.44 (m, 6H), 7.16 (d, *J* = 8.2 Hz, 3H), 6.88 (d, *J* = 8.4 Hz, 4H), 6.80 (d, *J* = 8.2 Hz, 2H), 3.82 (s, 6H). **^13^C NMR:** (126 MHz, CDCl_3_) δ 169.4, 160.1, 153.3, 150.9, 135.4, 133.4, 130.1, 128.0, 127.0, 126.4, 126.0, 125.4, 123.9, 120.0, 118.5, 114.8, 114.2, 91.4, 87.0, 81.9, 55.4. **HRMS** (ESI) *m*/*z*: [M + K]^+^ calculated for C_38_H_24_O_5_K; 599.1255, found 599.1255.


**Synthesis of 3’,6’-bis((4-(dimethylamino)phenyl)ethynyl)-3H-spiro[isobenzofuran-1,9’-xanthen]-3-one (Dye 2)**


The product was dried over anhydrous sodium sulfate, filtered, and concentrated under reduced pressure. The crude was loaded onto silica gel and purified by column chromatography using Hexanes:ethyl acetate (3:2) mixture to give a yellow-colored solid at 70% yield. **^1^H NMR:** (500 MHz, CDCl_3_) δ ^1^H NMR (500 MHz, CDCl_3_) δ 8.05 (d, *J* = 7.4 Hz, 1H), 7.65 (dtd, *J* = 22.1, 7.4, 1.2 Hz, 2H), 7.45–7.39 (m, 6H), 7.20–7.11 (m, 4H), 6.77 (d, *J* = 8.2 Hz, 2H), 6.66 (d, *J* = 8.9 Hz, 4H), 3.00 (s, 12H). **^13^C NMR:** (126 MHz, CDCl_3_) δ 169.5, 153.4, 151.0, 150.5, 135.4, 133.1, 130.0, 127.9, 127.0, 126.8, 126.2, 125.4, 124.0, 120.0, 117.9, 111.9, 109.3, 92.9, 86.5, 82.2, 40.3. **HRMS** (ESI) *m*/*z*: [M + K]^+^ calculated for C_40_H_30_N_2_O_3_K; 625.1888, found 625.1887.


**Synthesis of 3’,6’-bis((4-(diphenylamino)phenyl)ethynyl)-3H-spiro[isobenzofuran-1,9’-xanthen]-3-one (Dye 3)**


The product was dried over anhydrous sodium sulfate, filtered, and concentrated under reduced pressure. The crude was loaded onto silica gel and purified by column chromatography using Hexanes:ethyl acetate (75:25) mixture to give a yellow-colored solid at 85% yield. **^1^H NMR:** (500 MHz, CDCl_3_) δ 8.08 (d, *J* = 7.4 Hz, 1H), 7.67 (dt, *J* = 18.6, 7.4 Hz, 2H), 7.47 (s, 2H), 7.40 (d, *J* = 8.3 Hz, 4H), 7.30 (t, *J* = 7.7 Hz, 8H), 7.19 (d, *J* = 7.6 Hz, 3H), 7.14 (d, *J* = 7.9 Hz, 8H), 7.09 (t, *J* = 7.4 Hz, 4H), 7.03 (d, *J* = 8.3 Hz, 4H), 6.82 (d, *J* = 8.2 Hz, 2H). **^13^C NMR:** (126 MHz, CDCl_3_) δ 169.4, 153.3, 150.9, 148.44, 147.1, 135.4, 132.8, 130.1, 129.5, 127.9, 127.0, 126.4, 126.0, 125.4, 125.2, 123.9, 123.8, 122.1, 119.9, 118.4, 115.3, 91.8, 87.5, 81.9. **HRMS** (ESI) *m*/*z*: [M + K]^+^ calculated for C_60_H_38_N_2_O_3_K; 873.2514 found 873.2514


**Synthesis of 3’,6’-bis((4-(9H-carbazol-9-yl)phenyl)ethynyl)-3H-spiro[isobenzofuran-1,9’-xanthen]-3-one (Dye 4)**


The product was dried over anhydrous sodium sulfate, filtered, and concentrated under reduced pressure. The crude was loaded onto silica gel and purified by column chromatography using Hexanes:ethyl acetate (75:25) mixture to give a yellow-colored solid at 60% yield. **^1^H NMR:** (500 MHz, CDCl_3_) δ 8.17 (d, *J* = 7.8 Hz, 4H), 8.13–8.09 (d, 1H), 7.85–7.76 (d, 4H), 7.74–7.65 (m, 2H), 7.64–7.60 (d, 4H), 7.57 (s, 2H), 7.50–7.42 (m, 8H), 7.33 (t, *J* = 7.2 Hz, 4H), 7.28 (dd, *J* = 8.2, 1.6 Hz, 2H), 7.20 (d, *J* = 7.4 Hz, 1H), 6.90 (d, *J* = 8.1 Hz, 2H). **^13^C NMR:** (126 MHz, CDCl_3_) δ 169.4, 153.3, 150.9, 140.6, 138.2, 135.5, 133.4, 130.3, 128.1, 127.3, 127.0, 126.2, 126.0 125.8, 125.5, 123.9, 123.7, 121.6, 120.5, 120.4, 120.3, 119.1, 109.8, 90.6, 89.0, 81.7. **HRMS** (ESI) *m*/*z*: [M + K]^+^ calculated for C_60_H_34_N_2_O_3_K; 869.2201, found 869.2216


**General procedure for synthesis of Dyes 6–8**


Fluorescein ditriflate (1 equiv., 100 mg), 4-methoxyphenylboronic acid (3 equiv.), KOAC (6 equiv.), and Pd(PPh_3_)_4_ (6 mol%) were added into a flame-dried microwave reaction vial in a glovebox under nitrogen atmosphere. The microwave vial was sealed and removed from the glovebox. Then a mixture of 1,4-dioxane: water (1:2, 3 mL) was added, and the reaction mixture was heated at 100 °C for 12 h. The reaction mixture was allowed to cool to room temperature. This was followed by the addition of water (25 mL) and extraction with CH_2_Cl_2_ (3 × 20 mL). The combined organic layer was dried over anhydrous Na_2_SO_4_, filtered, and concentrated under reduced pressure.


**3′,6′-bis(4-methoxyphenyl)-3H-spiro[isobenzofuran-1,9’-xanthen]-3-one (Dye 6)**


The crude product was purified by column chromatography on silica gel with a Hexanes:ethyl acetate (65:35) mixture to give a colorless solid at 85% yield. ^1^H NMR: (500 MHz, CDCl_3_) δ 8.08 (d, *J* = 7.4 Hz, 1H), 7.74–7.62 (m, 2H), 7.56 (d, *J* = 8.9 Hz, 4H), 7.51 (d, *J* = 1.8 Hz, 2H), 7.26–7.22 (m, 3H), 6.99 (d, *J* = 8.7 Hz, 4H), 6.89 (d, *J* = 8.2 Hz, 2H), 3.86 (s, 6H). ^13^C NMR: (126 MHz, CDCl_3_) δ 169.6, 159.9, 153.6, 151.7, 143.6, 135.3, 132.15, 129.9, 128.5, 128.4, 126.5, 125.30, 124.1, 122.4, 117.1, 114.9, 114.5, 82.7, 55.5. **HRMS** (ESI) *m*/*z*: [M + K]^+^ calculated for C_34_H_24_O_5_K; 551.1255, found 551.1255.


**Synthesis of 3’,6’-bis(4-(dimethylamino)phenyl)-3H-spiro[isobenzofuran-1,9’-xanthen]-3-one (Dye 7)**


The crude product was purified by column chromatography on silica gel with a Hexanes:ethyl acetate (3:2) mixture to give a yellow solid at 78% yield. ^1^H NMR: (500 MHz, CDCl_3_) δ 8.09 (d, *J* = 7.5 Hz, 1H), 7.67 (dt, *J* = 21.8, 7.3 Hz, 2H), 7.56 (d, *J* = 8.3 Hz, 4H), 7.53 (d, *J* = 1.8 Hz, 2H), 7.30–7.22 (m, 3H), 6.87 (d, *J* = 8.2 Hz, 2H), 6.81 (d, *J* = 8.5 Hz, 4H), 3.02 (s, 12H). ^13^C NMR: (126 MHz, CDCl_3_) δ 169.7, 153.6, 151.8, 150.5, 143.9, 135.2, 129.8, 128.3, 127.9, 127.3, 126.6, 125.2, 124.1, 121.8, 116.3, 114.1, 112.7, 83.1, 40.5. **HRMS** (ESI) *m*/*z*: [M + K]^+^ calculated for C_36_H_30_N_2_O_3_K; 577.1887, found 577.1888.


**Synthesis of 3’,6’-bis(4-(diphenylamino)phenyl)-3H-spiro[isobenzofuran-1,9’-xanthen]-3-one (Dye 8)**


The crude was purified by column chromatography on silica gel with a Hexanes:ethyl acetate (8:2) mixture to give a yellow solid at 80% yield. ^1^H NMR: (500 MHz, CDCl_3_) δ 8.06 (d, *J* = 7.4 Hz, 1H), 7.71–7.58 (m, 2H), 7.49 (s, 2H), 7.45 (d, *J* = 8.7 Hz, 4H), 7.30–7.18 (m, 11H), 7.12 (dd, *J* = 8.2, 4.9 Hz, 12H), 7.03 (t, *J* = 7.4 Hz, 4H), 6.86 (d, *J* = 8.4 Hz, 2H). ^13^C NMR: (126 MHz, CDCl_3_) δ 169.6, 153.5, 151.7, 148.1, 147.5, 143.5, 135.3, 133.1, 129.9, 129.5, 129.1, 128.5, 127.9, 126.5, 125.3, 124.8, 124.1, 123.4, 123.4, 122.9, 122.2, 117.1, 114.8, 82.7. **HRMS** (ESI) *m*/*z*: [M + H]^+^ calculated for C_56_H_38_N_2_O_3_H; 787.2955, found 787.2966.


**Analysis of the ring opening of each probe using different equivalents of TFA**


Four milliliters of a 100 μM solution of each probe in chloroform was prepared, and different equivalents of TFA were added. The solutions were allowed to sit, and the absorption spectra were recorded after 30 min and 24 h.


**Computational Methods**


Density functional theory (DFT) with the Minnesota density functional M06-2X [58,59,60] was applied in the present investigation. The basis set used was the Standard triple zeta basis set augmented by polarization functions, and the diffuse functions—6-311G(d,p) was used [61,62,63]. The Barone–Tomasi polarizable continuum model (PCM) [64] with the recommended dielectric constant of dichloromethane DCM (*e* = 8.93) was applied to simulate the solvated environment of an aqueous solution. The ground state geometries of the study’s models were fully optimized using the above-mentioned theoretical level. The force constants were determined analytically in the analysis of harmonic vibrational frequencies for all of the complexes. The analysis of the data revealed that all considered structures were minimum energy structures.

Time-dependent density functional theory (TDDFT) has been developed for theoretical studies of excitation energies, absorption wavelengths, and oscillator strengths [65]. TDDFT is widely applied in optical investigations of both large and medium size molecules. The combination of DFT and TDDFT, as used in the current study, can provide efficient and reasonably accurate evaluations of excited state properties. The GAUSSIAN 09 system of DFT programs [66] was used for all computations.

## 4. Conclusions

We prepared donor-acceptor-donor far-red to NIR xanthene-based dyes with an alkyne spacer to determine the effect of the spacer and donors on the absorption spectra and the molecular switching (ring opening) process of the dyes. The dyes were prepared using a short synthetic route, and the Sonogashira reaction was used to connect the alkyne spacer containing the 4-anisole, N,N-dimethylaniline, N,N-diphenylaniline, and carbazole donors to the xanthene core. The trend in the abs λ_max_ of the dye was controlled by the ET strength of the donors experimentally; however, the calculations predicted one difference in the trend pertaining to Dye 2 containing the N,N-dimentylaniline donor. The K_eq_ constants of the opened and closed forms of the dyes can be determined by the π-donating strength of the donors. Finally, the alkyne spacer was shown to extend the conjugation length of the dyes, thus increasing the abs λ_max_ of the dyes, and the equilibrium constants of the dyes seem to increase without the alkyne spacer.

## Data Availability

Not applicable.

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
