# Peer review of "Investigating the Effects of Donors and Alkyne Spacer on the Properties of Donor-Acceptor-Donor Xanthene-Based Dyes"

_molecules, 2023, doi:10.3390/molecules28134929_

Round 1

Reviewer 1 Report

The authors present the synthesis and investigation of donor-acceptor-donor far-red to NIR xanthene-based dyes with an alkyne spacer to determine the effect of the spacer and donors on the absorption spectra. The subject is of interest for the journal readers, and the paper presents some valuable data. It was a real pleasure reading the paper. Nevertheless, I suggest the publication after addressing the following points:

1.       The presentation of the Dye 5 is questionable. Although only a 10% yield is obtained, there must be enough for UV-Vis analysis since very diluted solutions are required for the analysis. This also refers to the characterization of the particular dye.

2.       Figure 1 is unnecessary since these donor groups are also presented in Scheme 1

3.       Lines 166-169 are redundant since no information is obtained

4.       The large deviation of the calculated and experimental UV-Vis absorption maxima should be explained.

5.       Figure 2 does not have sufficient information. Namely, the spectra present UV-Vis absorption in DCM, but with addition of TFA, which is not indicated.

6.       Line 230- the text refers to the synthesis of dyes 6-8 to scheme 2, but there is no scheme 2 in the manuscript.

7.       The characterization of the compound 4, D4, compound 5 and D5 should be supplemented. Namely, if they are known in the literature, they should be referenced, or otherwise, the characterization should include at least 13C NMR.

8.       The characterization of all compounds is missing melting points and FTIR spectra. This is custom practice for reporting of new compounds.

9.       Figures of Dyes 2, 6 and 8 in Supplementary material are not adequate. Please correct.

10.   Some typing errors should be corrected:

L109 nitrogen-based donors – donor is also oxygen-based anisole, please add

L117 donors instead of doners

L266, L343, L397 – it seems that the text is missing

Reviewer 2 Report

Manuscript Molecules-2406525 reports on the synthesis and characterization of novel dyes based on a xanthene core symmetrically linked through ethynyl bridge to several donor moieties. The authors performed an optical characterization on the opened form of the dyes, obtained by treatment with TFA, and made a computational analysis focusing on the relative stability of closed and opened form of the dyes. Finally, they compared optical properties of the first set of the dyes with a second one where the donor moieties are directly linked to the xanthene core.

I think the presented materials are potentially interesting but some aspects in the manuscript need to be clarified and investigated more into detail. In the following, some points that should be improved:

-        Regarding the dye D5, the authors wrote it was obtained in poor yield (10 %) and they cannot perform optical characterization (why? The amount of material needed for optical characterization is not huge); anyway, they did not present any details about the synthesis or NMR spectrum. My suggestion is to completely remove D5 from the discussion.

-        The optical propertied of D2 (lmax at 515 nm) appear to be completely unexpected by looking at the properties of the analogous VIX3 dye, previously reported by the same authors (lmax at 875 nm) considering the very slight structure difference between the two dyes. In addition, also the computational analysis suggests for D2 absorption at wavelengths longer than those experimentally found. The author should try to give an explanation of this surprising behaviour.

-        The authors should present the optical spectra of the closed form of the dyes; these spectra could help to visualize the percentage of dyes that effectively opened after TFA treatment. Moreover, by knowing the variation in dye’s closed form during the TFA process, could probably provide a way to determine the equilibrium constant and correlating it with the calculated ones.

-        In Supporting information, the captions of Fig. S4 is not clear: I think that the straight and dashed lines in the Figure refer, respectively, to 24h and 30 min TFA treatment, but the author should clarify. Anyway, unlike what the authors wrote in the manuscript, some difference between 30min and 24h treatment are evident, at least for the lower amount of TFA equivalents.

-        I think it was a formatting error, but figures reporting NMR for dyes D2-3-6-8 are not legible and should be replaced.

-        The authors should add HRMS spectra of the dyes

-        Minor revision: page 6 line 200: D1 and D5 have the biggest K values and not the smallest as written by the authors; page 10  line 388: general procedure for the synthesis of…the authors should add dyes 6-8.

On the basis of the previous argumentations, I do not think the manuscript is publishable in its present form. I would be happy to change my opinion after the suggested major revisions will be made.

Author Response

Please check the new attachment.

Round 2

Reviewer 1 Report

All remarks have been answered. The paper is suitable for the publication.

Reviewer 2 Report

The authors did not provide a point-to-point response to my comments; I think that this is probably a cut and paste error in the redaction of the new cover letter because I can see they replied to the second reviewer. Considering that in the revised version of the manuscript they did not entirely reply to my suggestions I would like to have the corrected cover letter before giving my opinion on the possible publication of the manuscript. 

English quality is fine

Round 3

Reviewer 2 Report

The authors replied to all my remarks. The revised version of the manuscript is now suitable for publication.